# Endothelial glycocalyx and cardio-renal risk factors in type 1 diabetes

**Elisabeth Buur Stougaard**[1]\*, **Signe Abitz Winther**[1], **Hanan Amadid**[2], **Marie Frimodt-Møller**[1], **Frederik Persson**[1], **Tine Willum Hansen**[1], **Peter Rossing**[1,3]

**1** Complication Research, Steno Diabetes Center Copenhagen, Gentofte, Denmark, **2** Clinical Epidemiology, Steno Diabetes Center Copenhagen, Gentofte, Capital Region, Denmark, **3** Department of Clinical Medicine, University of Copenhagen, Copenhagen, Denmark

\* elisabeth.buur.stougaard@regionh.dk

**Data Availability Statement:** All relevant data are within the paper and its Supporting Information files.

## Abstract

### Background

Glycocalyx lines the inner surface of the capillary endothelium. Capillaroscopy enables visualization of the sublingual capillaries and measurement of the Perfused Boundary Region (PBR) as an estimate of the glycocalyx. Novel software enables assessment of the PBR estimated at a fixed high flow level (PBR-hf) and an overall microvascular assessment by the MicroVascular Health Score (MVHS). Damaged glycocalyx may represent microvascular damage in diabetes and assessment of its dimension might improve early cardio-renal risk stratification.

### Aim

To assess the associations between PBR, PBR-hf and MVHS and cardio-renal risk factors in persons with type 1 diabetes (T1D); and to compare these dimensions in persons with T1D and controls.

### Methods

Cross-sectional study including 161 persons with T1D stratified according to level of albuminuria and 50 healthy controls. The PBR, PBR-hf and MVHS were assessed by the Glyco-Check device (valid measurements were available in 136 (84.5%) with T1D and in all the controls). Higher PBR and PBR-hf indicate smaller glycocalyx width. Lower MVHS represents a worse microvascular health.

### Results

There were no associations between PBR, PBR-hf or MVHS and the cardio-renal risk factors in persons with T1D, except for higher PBR-hf and lower MVHS in females (p = 0.01 for both). There was no difference in PBR, PBR-hf or MVHS in persons with normo-, micro- or macroalbuminuria. The PBR was higher (2.20±0.30 vs. 2.03±0.18µm; p<0.001) and MVHS lower (3.15±1.25 vs. 3.53±0.86µm; p = 0.02) in persons with T1D compared to controls (p≤0.02). After adjustment for cardio-renal risk factors the difference in PBR remained significant (p = 0.001).

**Funding:** The project is funded by the Novo Nordisk Foundation (grant number NNF14OC0013659; 'PROTON: PeRsOnalising Treatment Of diabetic Nephropathy' and by the Innovation Fund Denmark (grant number 5016-00150B). www.novonordiskfonden.dk www.innovationsfonden.dk The funders had no role in study design, data collection and analysis, decision to publish, or preparation of the manuscript.

## Conclusions

The endothelial glycocalyx dimension was impaired in persons with T1D compared to controls. We found no association between the endothelial glycocalyx dimension and the level of albuminuria or cardio-renal risk factors among persons with T1D. The use of the Glyco-Check device in T1D may not contribute to cardio-renal risk stratification.

## Introduction

Presence of microvascular disease is strongly associated with increased cardiovascular risk and is a major contributor to the increased morbidity and mortality observed in type 1 diabetes [1–3]. Diabetic kidney disease is a frequent and severe microvascular complication to type 1 diabetes and the symptoms are often vague or non-existing during the early stages. Once symptoms are present, the extent of the microvascular damage might be severe. Thus, there is a strong need to identify individuals in highest risk of microvascular complications to prevent development or progression. Estimation of the endothelial glycocalyx dimension might be a clinical tool for early risk stratification of microvascular damage in persons with type 1 diabetes [4]. If modifiable by intervention, glycocalyx dimensions could also be an endpoint in intervention studies to assess early treatment benefits.

The glycocalyx is a glycoprotein, gel-like layer that lines and protects the inner surface of the capillary endothelium. Loss of or a damaged endothelial glycocalyx may represent initial microvascular damage [5]. The glycocalyx consists of a luminal layer, which allows penetration of the blood cells. The distance between the median and the outer edge of this perfused lumen is called the Perfused Boundary Region (PBR) **(Fig 1)** as published by Eickhoff [6]. The PBR reflects the thickness of the endothelial glycocalyx, since loss of its integrity allows deeper penetration of the red blood cells into the gel-like layer covering the endothelial lining. A higher PBR indicates thinner glycocalyx and is related to presence of early stages of atherosclerosis, albuminuria and other diabetic complications [7–9]. Several methods have been developed to study the glycocalyx dimensions in humans [10]. Some have been more difficult than others requiring invasive procedures [10, 11]. The use of side stream dark field (SDF) imaging to automatically calculate the PBR from short video recordings of the sublingual microcirculation, is a fast and non-invasive method to estimate the dimensions of the endothelial glycocalyx [12, 13]. It is, however, unclear what the potential of this method is for cardio-renal risk stratification. Several studies have investigated the associations between the PBR, measured with SDF imaging, and the presence of risk factors for cardiovascular and renal disease in different populations. Results have been inconsistent. Some studies could not demonstrate an association between glycocalyx size and cardiovascular or renal disease [14, 15] and concluded that the technique might not contribute to cardiovascular risk stratification [16]. Whereas a higher PBR in first degree relatives of persons with premature coronary artery disease as compared to healthy controls has been described [17]. None of these studies have been performed in a cohort solely of persons with type 1 diabetes as in our study and with a sample size as large as ours.

An updated software of the GlycoCheck system enables assessment of the PBR at a fixed high flow level (PBR-hf) and calculation of a MicroVascular Health Score (MVHS). The importance of these new measures has only been sparsely investigated and, to our knowledge, described only in one study focusing on dengue and other febrile illnesses [18]. Thus, the impact of these measures in persons with diabetes is unknown.

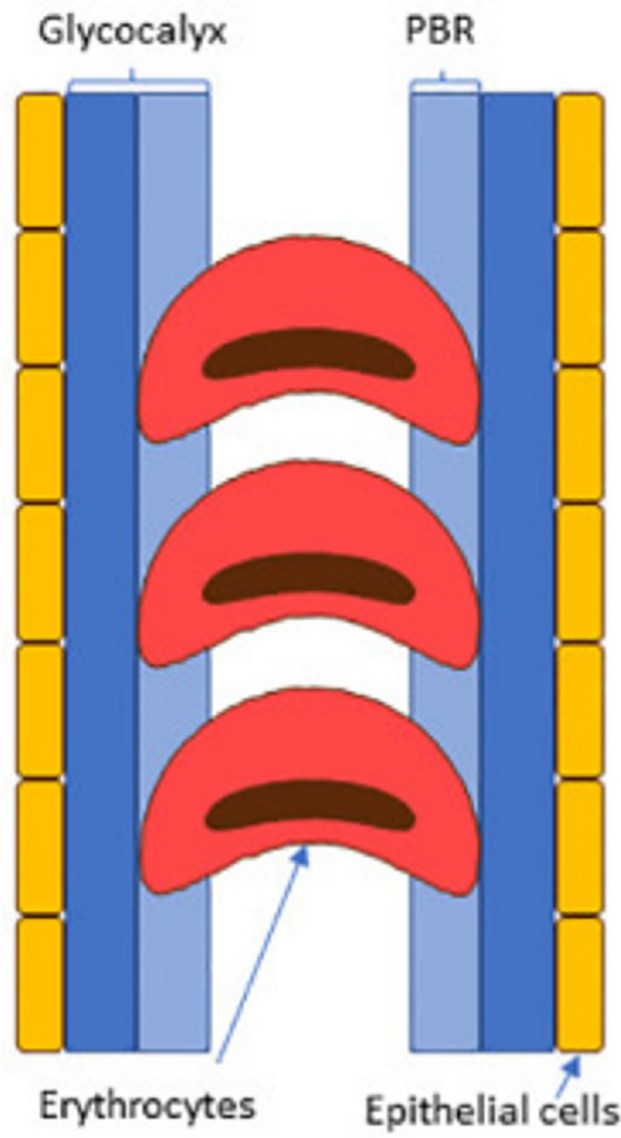

**Fig 1. Schematic representation of the Perfused Boundary Region (PBR) [6].**

In the present study, we investigated the associations between these measures of endothelial glycocalyx dimension (PBR, PBR-hf and MVHS) and cardio-renal risk factors and levels of albuminuria in persons with type 1 diabetes. Moreover, we compared these measurements in persons with type 1 diabetes and healthy controls.

## Materials and methods

### Study population

A cross-sectional sample of 161 individuals with type 1 diabetes and 50 healthy control individuals without diabetes were recruited during April 2016 to December 2017 for a study that aimed to examine the microbiome and glycocalyx in type 1 diabetes [19]. Subjects with type 1

diabetes were recruited from Steno Diabetes Center Copenhagen and identified through our electronic medical records. Potentially suitable participants were in writing given the offer to participate. Non-responders were contacted by telephone and given a renewed offer to participate in the study. Healthy control subjects were recruited from newspaper advertisement, where they were encouraged to contact the responsible investigator for more information. If after the telephone conversation the subject was still interested, detailed information was sent for further review. The included individuals with type 1 diabetes were >18 years of age and diagnosed according to the WHO criteria. Main exclusion criteria were: (1) non-diabetic kidney disease; (2) renal failure (estimated glomerular filtration rate (eGFR) <15 ml/min/1.73 m$^2$), dialysis or kidney transplantation; and (3) change in renin–angiotensin–aldosterone system (RAAS)-blocking treatment during the month prior to study inclusion. The control group was healthy volunteers by self-report. None of the healthy controls took any prescription medication. Participants included in the current study were recruited from a pool of approximately 3,500 persons with type 1 diabetes attending the outpatient clinic at Steno Diabetes Center Copenhagen. Thus, almost 7% of persons followed-up at Steno Diabetes Center Copenhagen were investigated, representing a broad segment of the Steno population, which covers an unselected population of adults with type 1 diabetes in the capital region of Denmark. The individuals with type 1 diabetes were stratified into three groups of albuminuria based on the highest urine albumin/creatinine ratio (UACR) level measured at the study visit or documented previously in two out of three consecutive urine samples within 1 year (as albumin content in 24 hour urine samples [UAER] or UACR). The three albuminuria groups consisted of 50 persons with normoalbuminuria (<30 mg/24 h or mg/g), 50 with microalbuminuria (30–299 mg/24 h or mg/g) and 61 with macroalbuminuria (≥300 mg/24 h or mg/g). Participants classified with normoalbuminuria did not have any recorded history of micro- or macroalbuminuria. For the macroalbuminuria group, at least 30 individuals were selected based on concurrent eGFR <60 ml/min/1.73 m$^2$, to ensure an as broad representation of participants in this group as possible.

Study participants were recruited to ensure equal distribution of sex and similar mean age in the three albuminuria groups and the healthy controls. The study was conducted in accordance with the Declaration of Helsinki and approved by the Ethics Committee of the Danish Capital Region (protocol H-15018107). All participants gave written informed consent.

### Bioclinical measures

The bioclinical measures have previously been described in detail [19]. In short, laboratory measures included HbA$_{1c}$, lipid profile and plasma creatinine, which were measured by standard methods [19] and eGFR was calculated using the Chronic Kidney Disease Epidemiology Collaboration (CKD-EPI) equation from standardized serum creatinine [20]. UACR was measured in three consecutive morning urine samples by an enzyme immunoassay and the geometric mean was calculated for each participant. Twenty-four-hour blood pressure was recorded with a validated device programmed to measure blood pressure every 15 minutes between 07:00 hours and 22:00 hours and every 30 minutes between 22:00 hours and 07:00 hours (TM2430, Takeda, Japan) [21]. Demographic characteristics, body mass index (kg/m$^2$), smoking status and a detailed medical history along with information on medical treatment was obtained by questionnaires and from medical records. Smoking was classified as current (one or more cigarettes/cigars/pipes a day) or non-smoking.

Retinopathy status was obtained from medical records. All participants attending the outpatient clinic at Steno Diabetes Center Copenhagen have regular ophthalmology examinations (approximately every 1–2 years) where retinal photography is taken through a dilated pupil by

certified eye nurses. Retinopathy was graded as nil, presence of or historical simplex or proliferative, based on the worst eye.

## Measurements of endothelial glycocalyx dimensions

The GlycoCheck system has previously been described in detail [22]. In short, the system consists of a small handheld video camera connected to a computer with the GlykoCheck software. The microscopy was performed with the participant sitting on a chair. The sublingual capillaries were visualized using an SDF video microscope (Capiscope handheld, KK Research technology Ltd), which uses green light emitting diodes to detect the haemoglobin of passing red blood cells (RBC), and the dimensions of the glycocalyx are then estimated with the integrated software (GlycoCheck™, Maastricht, The Netherlands). During the video recording, the software automatically detects valid blood capillary segments for recordings and performs measurements of PBR in segments automatically positioned every 10 μm along the capillary. Capillaries with a diameter between 5 and 25 μm are automatically identified. Data acquisition automatically starts when image quality is within acceptable range and are automatically stopped when data on a minimum number of 3000 measurement sites have been obtained. For each vascular segment, the dynamic lateral position of RBCs (per RBC column width) is then calculated. The cumulative distribution is calculated from the intensity profiles of the dispersal of RBC column widths and the median RBC column width is determined. The PBR is then defined as the distance between RBC column width and perfused diameter. Next, the calculated PBR values, classified along with their corresponding RBC column width between 5–25 μm, are averaged and a single PBR value is provided for each person. As mentioned previously, the PBR reflects the thickness of the endothelial glycocalyx. Higher PBR indicates thinner glycocalyx [22]. The software also enables assessment of the PBR estimated at a fixed high flow level (PBR-hf), which is suggested to be more accurate than measurements in capillaries without flow [18]. A higher PBR-hf reflects a more degraded glycocalyx. In addition, the software calculates the MVHS using the PBR-hf plus parameters representing the RBC filling and valid vessel density (the subset of vessels with enough red cells for tissue perfusion). Lower values of MVHS is suggested to reflect an overall worse microvascular health [18].

Five measurements were performed in each participant, and the mean of valid measurements was calculated. Measurements were performed in the morning after three hours of fasting which included food intake, all beverages and smoking [6].

Investigators were trained to handle the GlycoCheck camera and system by the manufacturer.

Due to technical issues (one of the four diodes in the camera was broken) the video quality was poor in a period during the study and some of the recordings therefore had to be deleted. The video quality was visually judged by a technician from the GlycoCheck company blinded to all other data in the study. The measurements were not valid in 25 (15.5%) of the persons with type 1 diabetes (11 with normo-, 5 with micro- and 9 with macroalbuminuria). All the healthy controls had valid measurement.

## Statistical analyses

The sample size was originally estimated to test for diversity in the microbiome in participants with diabetes compared to healthy controls, as already published [19]. Glycocalyx measurement was a prespecified secondary endpoint, and we performed a post hoc sample size calculation (using the power statement implemented in the SAS software, version 9.3) also for the analysis in the present paper. We considered a difference of 0.2 μm in PBR to be relevant with a variance of 0.3 μm [16] and calculated that at least 21 participants should be included in each albuminuria group to obtain a significance level of 0.05 and a power of 80%. Thus, the numbers of participants in this study exceeded the number needed.

Continuous variables are presented as mean ± standard derivation (SD) if normal distributed, and the non-normal distributed variable (UACR) is presented as median (interquartile range (IQR)) and log-transformed before analyses to achieve normal distribution. Categorical variables are reported as numbers (%).

The participants with diabetes were stratified by PBR into quartiles. One Way Analysis of variance (ANOVA) and χ2-statistic were used to compare means and proportions across the quartiles.

Unadjusted linear regression models were applied to examine the association between PBR, PBR-hf and MVHS and the cardio-renal risk factors.

ANOVA was applied when comparing levels of PBR, PBR-hf and MVHS between the three albuminuria groups (normo-, micro- and macroalbuminuria) and in four eGFR groups ($> 90$, 60–90, 45–59 and $< 45$ ml/min/1.73m$^2$) and Welch Two Sample t-test was applied when comparing levels of PBR, PBR-hf and MVHS between persons with diabetes and healthy controls. Analysis of covariance (ANCOVA) was performed to control for the effects of covariates on all analyses. The covariates adjusted for in the ANOVA and ANCOVA models included age, sex, 24-hour systolic blood pressure, eGFR (except for analyses comparing the four eGFR groups), UACR (except for analyses comparing the three albuminuria groups), HbA$_{1c}$ (except for analyses comparing persons with diabetes and healthy controls), LDL-cholesterol, smoking and treatment with RAAS-inhibitors and statins.

In all analyses, the model assumptions were ascertained. A two-tailed p-value of $<0.05$ was considered significant. Statistical analysis was performed using R software (version R i386 3.6.1)

## Results

### Clinical characteristics

The clinical characteristics of the people with type 1 diabetes and the healthy controls are presented in Table 1.

Among the 136 individuals with type 1 diabetes, 57 (42%) were women, the mean ± SD age was 60±10 years and diabetes duration was 40±14 years. Median (IQR) UACR was 17 (5–113) mg/g with 39 (29%), 45 (33%) and 52 (38%) having normoalbuminuria, microalbuminuria and macroalbuminuria respectively and 39 (29%) had a history of cardiovascular disease. Among the 50 healthy controls, 23 (46%) were women and the mean age was 59±12 years. Table 1 also shows the clinical characteristics of the persons with type 1 diabetes divided into quartiles of PBR. There was no significant difference between any of the clinical characteristics across the quartiles (p≥0.13).

### Levels of PBR, PBR-hf and MVHS in people with type 1 diabetes and the healthy controls

The PBR was higher (2.20±0.30 vs. 2.03±0.18 μm; p<0.001) and MVHS was lower (3.15±1.25 vs. 3.53±0.86; p = 0.02) in persons with T1D as compared to the healthy controls (**Fig 2**). There was no difference in PBR-hf between the two groups (p = 0.13). After adjustment for cardio-renal risk factors the difference in PBR remained significant (p = 0.001), but significance was lost for MVHS (p = 0.19).

### Associations between PBR, PBR-hf and MVHS and cardio-renal risk factors in persons with type 1 diabetes

Table 2 shows the unadjusted associations between PBR, PBR-hf and MVHS and cardio-renal risk factors in the people with type 1 diabetes.

**Table 1. Clinical characteristics of healthy controls, persons with type 1 diabetes and across quartiles of PBR among the persons with type 1 diabetes.**

| | Controls | All diabetes individuals | PBR in quartiles among type 1 diabetes | | | | p-value |
|---|---|---|---|---|---|---|---|
| Range (μm) | | | 1.5-<1.98 | 1.98-<2.18 | 2.18-<2.39 | 2.39–3.21 | |
| Subjects (N) | 50 | 136 | 34 | 34 | 34 | 34 | |
| Age (years) | 59±12 | 60±10 | 58±9 | 60±10 | 59±9 | 62±12 | 0.41 |
| Female (%) | 23 (46) | 57 (42) | 10 (29) | 15 (44) | 16 (47) | 16 (47) | 0.39 |
| Body mass index (kg/m$^2$) | 24±3.3 | 26±4.4 | 28±4.5 | 26±5.2 | 25±4.1 | 26±3.4 | 0.13 |
| 24h systolic BP (mmHg) | 133±12 | 138±12 | 140±14 | 135±12 | 136±12 | 140±10 | 0.27 |
| 24h diastolic BP (mmHg) | 80±7 | 77±6 | 78±6 | 76±6 | 77±6 | 77±6 | 0.60 |
| eGFR (ml/min/1.73m$^2$) | 88±14 | 76±25 | 75±23 | 79±24 | 74±26 | 75±26 | 0.89 |
| UACR (mg/g) | 4 (3–5) | 17 (5–113) | 17 (4–180) | 19 (4–98) | 10 (5–52) | 21 (5–189) | 0.53 |
| HbA$_{1c}$ (mmol/mol) HbA$_{1c}$ (%) | 36±2.7 5.4 ±0.3 | 61±10.0 7.8±0.9 | 62±12.0 7.9±1.1 | 62±8.9 7.8±0.8 | 60±9.6 7.7±0.9 | 62±9.4 7.8±0.9 | 0.84 |
| LDL cholesterol (mmol/L) | 3.2±0.8 | 2.1±0.7 | 2.2±0.8 | 2.1±0.7 | 2.1±0.6 | 2.1±0.6 | 0.70 |
| Smokers (%) | 4 (8) | 18 (14) | 4 (12) | 6 (19) | 2 (6) | 6 (18) | 0.43 |
| Diabetes characteristics | | | | | | | |
| Diabetes duration (years) | - | 40±14 | 41±13 | 40±13 | 40±14 | 39±17 | 0.96 |
| History of cardiovascular disease (%) | - | 39 (29) | 10 (31) | 10 (29) | 9 (27) | 10 (29) | 0.99 |
| Retinopathy (no/simplex/proliferative) (%) | - | 28(21)/46(34)/61(45) | 6(18)/12(35)/16(47) | 9(27)/9(2)7/16(47) | 4(12)/14(42) /15(46) | 9(27)/11(32) /14(41) | 0.69 |
| Medication | | | | | | | |
| RAAS-blockers (%) | - | 30 (22) | 9 (27) | 8 (24) | 4 (12) | 9 (27) | 0.42 |
| Statins (%) | - | 38 (27) | 8 (24) | 12 (31) | 9 (27) | 9 (27) | 0.92 |
| Antihypertensives (%) | - | 66 (48) | 16 (47) | 15 (41) | 22 (65) | 13 (38) | 0.12 |
| Aspirin (%) | - | 70 (51) | 20 (59) | 17 (47) | 16 (47) | 17 (50) | 0.74 |

The data represent %, mean ± standard deviation or median (interquartile range). P for difference between quartiles was calculated using analysis of covariance for continuous variables and the $\chi^2$-test for categorical variables. UACR: urine albumin creatinine ratio. RAAS-blockers: renin-angiotensin-aldosterone system-blockers.

The cardio-renal risk factors included age, sex, body mass index, 24-hour systolic blood pressure, eGFR, UACR, HbA$_{1c}$, LDL cholesterol, current smoking, diabetes duration, presence of cardiovascular disease and retinopathy. No significant associations were demonstrated, except for a higher PBR-hf and a lower MVHS in females (p = 0.01 for both). The associations between sex and PBR-hf and MVHS remained significant after adjustment (p = 0.03 for both).

## Levels of PBR, PBR-hf and MVHS according to markers of kidney function

Table 3 shows the clinical characteristics of the participants categorized into normo-, micro- and macroalbuminuria.

The eGFR was lower and the UACR, diabetes duration and the numbers with a history of cardiovascular disease and presence of retinopathy were, as expected, higher with increasing albuminuria grouping (p<0.001 for all). We found no significant difference in level of PBR, PBR-hf or MVHS between persons with normo-, micro- or macroalbuminuria (**Fig 3**). Neither overall (p≥0.34) nor between any of the three albuminuria groups (p≥0.34).

There was no significant difference (p≥0.53) in level of PBR, PBR-hf or MVHS between persons with different levels of eGFR (> 90, 60–90, 45–59 and < 45 ml/min/1.73m$^2$) (**Fig 4**). The lack of associations between the three albuminuria groups and the four eGFR groups persisted after adjustment for other risk factors.

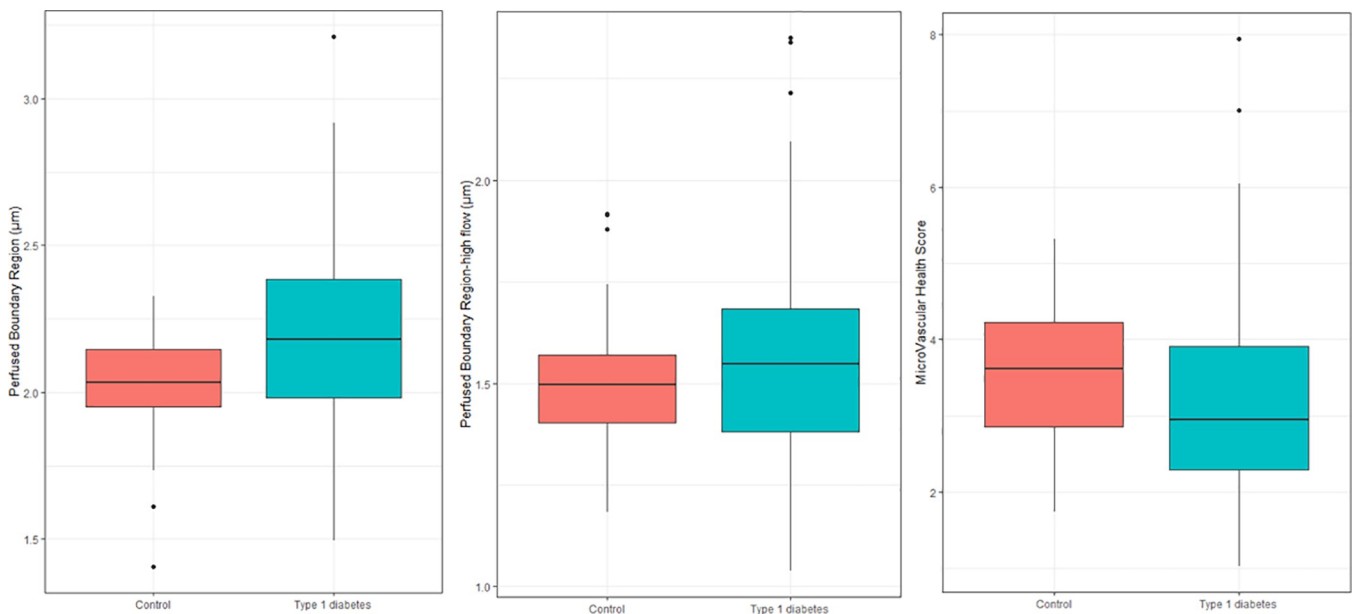

**Fig 2. Glycocalyx measurements within the group of healthy controls and type 1 diabetes.** Perfused Boundary Region (p<0.001), Perfused Boundary Region-high flow (p = 0.13) and MicroVascular Health Score (p = 0.02). P-values are for the unadjusted analyses.

## Discussion

In this study we applied the GlycoCheck system to evaluate the perfused boundary region (PBR) as an estimate of the endothelial glycocalyx, an estimate of the PBR at a fixed high flow level (PBR-hf) and an overall vascular assessment by the MicroVascular Health Score (MVHS) in persons with type 1 diabetes as compared to healthy controls. We demonstrated that the endothelial glycocalyx dimension was impaired in persons with type 1 diabetes as compared to healthy controls. We were not able to find associations between the endothelial glycocalyx

**Table 2. Unadjusted associations between measures of endothelial glycocalyx dimension and cardio-renal risk factors in the persons with type 1 diabetes (n = 136).**

|  | PBR | | PBR-hf | | MicroVascular Health Score | |
|---|---|---|---|---|---|---|
|  | β | P | β | P | β | P |
| Age | 0.04 | 0.17 | 0.01 | 0.70 | -0.10 | 0.39 |
| Female sex | 0.08 | 0.14 | 0.12 | **0.01** | -0.60 | **0.01** |
| Diabetes duration | -0.01 | 0.57 | -0.005 | 0.84 | 0.10 | 0.42 |
| Body mass index | -0.04 | 0.17 | -0.02 | 0.49 | 0.13 | 0.23 |
| 24h systolic BP | 0.003 | 0.93 | 0.01 | 0.82 | 0.16 | 0.19 |
| HbA1c | -0.01 | 0.62 | 0.003 | 0.91 | 0.07 | 0.53 |
| LDL cholesterol | -0.01 | 0.79 | -0.01 | 0.61 | 0.11 | 0.32 |
| eGFR | -0.02 | 0.34 | -0.01 | 0.71 | -0.002 | 0.99 |
| UACR | 0.02 | 0.49 | -0.03 | 0.18 | 0.19 | 0.11 |
| Current smoking | -0.03 | 0.69 | 0.004 | 0.96 | -0.22 | 0.52 |
| Retinopathy | -0.01 | 0.66 | -0.002 | 0.95 | 0.21 | 0.17 |
| History of cardiovascular disease | 0.02 | 0.69 | -0.01 | 0.81 | -0.16 | 0.54 |

The β-estimates represent standardized effect. PBR: Perfused Boundary Region; PBR-hf: Perfused Boundary Region at a fixed high flow level; UACR: urine albumin creatinine ratio.

**Table 3. Clinical characteristics of the participants categorized into normo-, micro- and macroalbuminuria.**

| | Normoalbuminuria | Microalbuminuria | Macroalbuminuria | p for normo- vs micro- vs macroalbuminuria |
|---|---|---|---|---|
| Subjects (N) | 39 | 45 | 52 | |
| Age (years) | 59±11 | 61±9 | 60±10 | 0.55 |
| Female (%) | 18 (46) | 19 (42) | 20 (39) | 0.76 |
| Body mass index (kg/m$^2$) | 26±4.2 | 26±4.8 | 27±4.1 | 0.60 |
| 24h systolic BP (mmHg) | 134±10 | 140±13 | 139±12 | 0.05 |
| 24h diastolic BP (mmHg) | 78±6 | 76±7 | 76±6 | 0.40 |
| eGFR (ml/min/1.73m$^2$) | 89±17 | 81±23 | 61±23 | <0.001 |
| UACR (mg/g) | 3 (2–5) | 13 (5–34) | 164 (54–416) | <0.001 |
| HbA$_{1c}$ (mmol/mol) HbA$_{1c}$ (%) | 60±8.4 7.6±0.8 | 60±6.5 7.6±0.6 | 64±13.0 8.0±1.1 | 0.08 |
| LDL cholesterol (mmol/L) | 2.3±0.6 | 2.1±0.6 | 2.1±0.7 | 0.36 |
| Smokers (%) | 3±8 | 8±18 | 7±14 | 0.45 |
| Diabetes characteristics | | | | |
| Diabetes duration (years) | 33±15 | 44±13 | 42±12 | <0.001 |
| History of cardiovascular disease (%) | 4 (11) | 10 (23) | 25 (49) | <0.001 |
| Retinopathy (no/simplex/proliferative) (%) | 18(46)/17(44)/4(10) | 7(16)/13(30)/24(55) | 3(6)/16(31)/33(64) | <0.001 |
| Medication | | | | |
| RAAS-blockers (%) | 20 (51) | 37 (84) | 47 (92) | <0.001 |
| Statins (%) | 16 (41) | 9 (20) | 11(22) | 0.06 |

Data represent numbers (%), mean ± standard deviation or median (interquartile range). P for difference between the albuminuria groups was calculated using analysis of covariance for continuous variables and the $\chi^2$-test for categorical variables. UACR: urine albumin creatinine ratio. RAAS-blockers: renin-angiotensin-aldosterone system-blockers.

dimension and cardio-renal risk factors, except for a higher PBR-hf and a lower MVHS in females, independent of other risk factors. We found no association between the endothelial glycocalyx dimension and the level of albuminuria or eGFR among persons with type 1 diabetes. The use of the GlycoCheck system may therefore not contribute to cardiovascular or renal risk stratification in persons with type 1 diabetes.

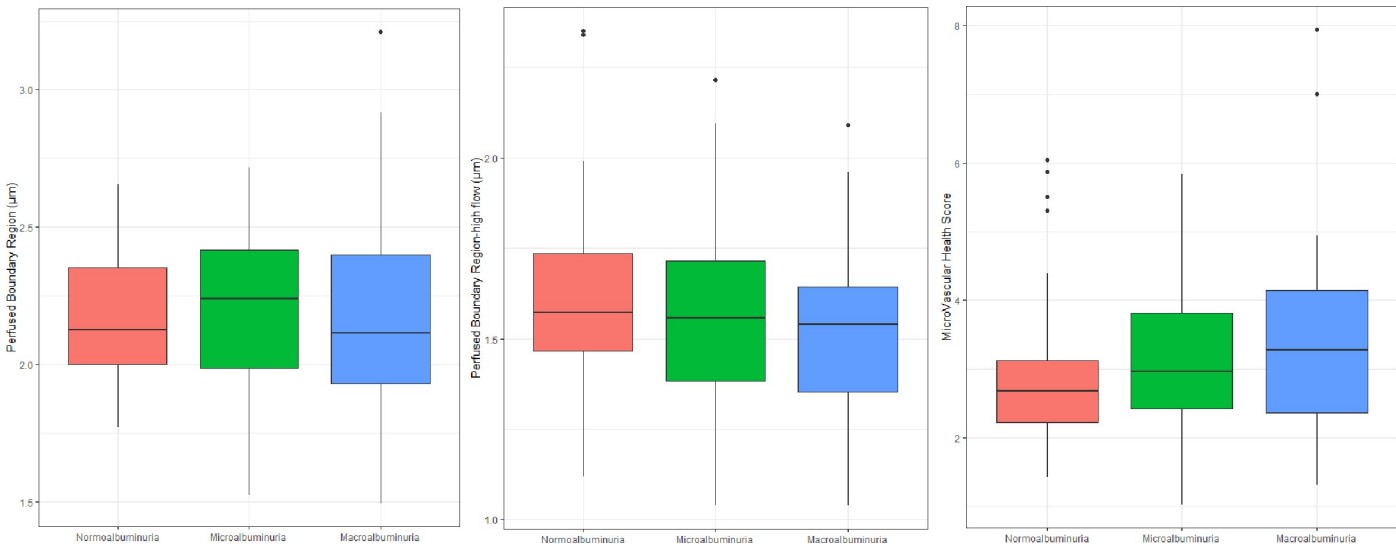

**Fig 3. Glycocalyx measurements in participants with type 1 diabetes and normo-, micro- and macroalbuminuria.** Perfused Boundary Region (p = 0.83), Perfused Boundary Region- high flow (p = 0.41) and MicroVascular Health Score (p = 0.34). P-values are for the unadjusted analyses.

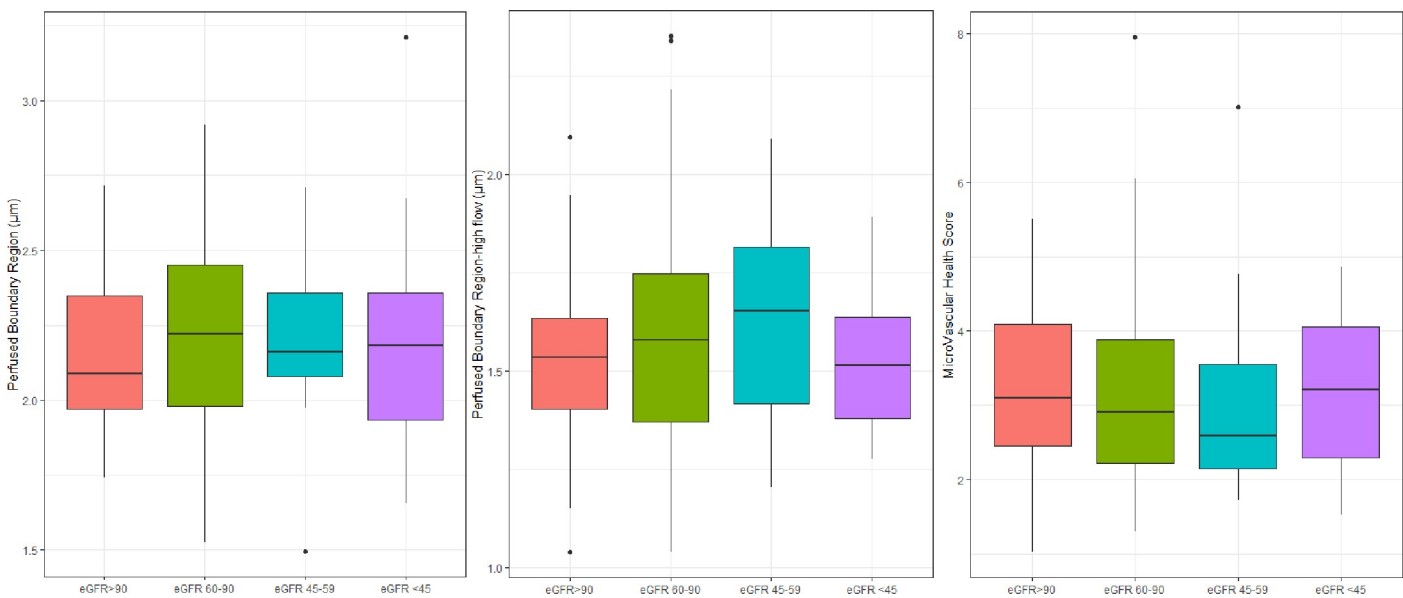

**Fig 4. Glycocalyx measurements in participants divided into groups according to eGFR level.** Perfused Boundary Region (p = 0.86), Perfused Boundary Region-high flow (p = 0.53) and MicroVascular Health Score (p = 0.999). P-values are for the unadjusted analyses.

The GlycoCheck system is an easy to use, non-invasive method for measurement of the glycocalyx dimension. The importance of the endothelial glycocalyx in various disease processes has become increasingly acknowledged and studying the size of the glycocalyx in type 1 diabetes has therefore also been of great interest. Damage to the glycocalyx may be the first sign of vascular damage progressing to microvascular complications leading to chronic kidney disease in diabetes. It has also been suggested that improvement of glycocalyx could be an early endpoint in intervention studies. However, it has been difficult to assess in clinical studies. This study is to our knowledge the first to examine the association between the endothelial glycocalyx dimension measured with the GlycoCheck system and cardio-renal risk factors and the level of albuminuria among persons with type 1 diabetes.

## Levels of PBR, PBR-hf and MVHS in persons with type 1 diabetes and healthy controls

The higher PBR in persons with diabetes as compared to healthy controls has also been reported in previous studies applying SDF imaging. In 2010 Broekhuizen et al. described reduced sublingual glycocalyx dimension in persons with type 2 diabetes compared with healthy controls [23], and later Groen et al. showed that older individuals with type 2 diabetes (n = 15, mean age 70 years) had a higher PBR than healthy young controls (n = 15, mean age 24 years), but similar PBR to that of healthy older controls (n = 15, mean age 70 years) [24]. According to type 1 diabetes, the study from Nieuwdorp et al. (applying orthogonal polarization spectral microscopy and a tracer method) showed that systemic glycocalyx volume was reduced in persons with long-standing type 1 diabetes compared with normoglycemic control subjects [9].

Not all studies have found an association between presence of diabetes and level of PBR. In the study by Amraoui et al., no relationship between PBR and presence of diabetes could be demonstrated [16] and Wadowski et al. found no difference in PBR dimensions between 36 persons with diabetes ((type 1: n = 20, type 2: n = 16) and 36 healthy controls [25].

The current hypothesis, that a hyperglycemic state reduces glycocalyx size, is supported by various studies examining the effect of different types of antidiabetic treatment on glycocalyx dimension. A study by Lambadiari et al. showed that intensified glycemic control in type 2 diabetes improved endothelial glycocalyx after one year of treatment with insulin or incretin-based therapy, suggesting that excessive hyperglycaemia might contribute to the loss of glycocalyx integrity [26]. This is consistent with findings from a randomized controlled study by Ikonomidis et al. investigating the effect of insulin, glucagon-like peptide-1 receptor agonists (GLP-1RA), sodium-glucose cotransporter-2 inhibitors (SGLT2i), and their combination on vascular and cardiac function of persons with type 2 diabetes. The study showed that the combined treatment with a GLP-1RA and a SGLT2i increased the endothelial glycocalyx thickness, as assessed by PBR [27]. Thus, these studies indicate that treatment with insulin, SGLT2i or incretin-based therapy could improve glycocalyx dimensions measured by use of the Glyco-Check system in type 2 diabetes. Therefore, the thickness of glycocalyx measured with the GlycoCheck system might be useful to monitor the effect of treatment at the individual level. It remains, although to be demonstrated if an improved glycocalyx will also translate into an improved prognosis for the individual person.

## Associations between endothelial glycocalyx dimensions and cardio-renal risk factors

In our study we found no association between the endothelial glycocalyx dimension and cardio-renal risk factors except for a higher PBR-hf and a lower MVHS in females. This is consistent with previous studies in other populations. A study by Amraoui et al. including persons visiting an outpatient clinic for vascular medicine and with different cardiovascular risk profiles, could not demonstrate any association between the endothelial glycocalyx dimension and known cardiovascular risk factors. Moreover, the PBR levels were similar among persons with and without cardiovascular disease, and in persons at high and low cardiovascular risk and in healthy controls. They therefore suggested that estimation of endothelial glycocalyx dimension by SDF imaging might not be useful for cardiovascular risk prediction [16]. Along similar lines, a recent study was unable to detect a difference in the PBR level between healthy controls, persons with chronic kidney disease, persons on dialysis, and in kidney transplant recipients [14]. On the contrary, Mulders et al. found that first degree relatives of persons with premature coronary artery disease were characterized by a higher PBR compared to healthy controls and concluded that sublingual capillary microvascular dysfunction might be useful for early risk prediction within families with premature coronary artery disease [17]. The findings were however, restricted to a population of families with premature coronary artery disease and the findings cannot be directly translated to other populations. A population based study by Valerio et al. [15] showed that persons with a PBR in the highest quartile were more likely to be female and diagnosed with diabetes after controlling for possible confounders including age, diastolic blood pressure and body mass index. But, the glycocalyx size was not associated with other cardiovascular risk factors or presence of cardiovascular disease. The authors did not analyze PBR as a continuous variable. Finally, a study by Wadowski et al. demonstrated that persons with type 1 diabetes and HbA$_{1c}$ levels $\geq$ 8% had a significantly higher PBR compared to person with HbA$_{1c}$ levels < 8% but could not demonstrate a similar association in persons with type 2 diabetes [25]. The study also showed an inverse correlation between PBR and creatinine, but microvascular parameters (PBR, red blood cells filling percentage, perfused and total capillary density) did not correlate with eGFR. We were also unable to show a correlation between endothelial glycocalyx dimension and eGFR [25].

The gender difference in PBR-hf and MVHS was quite unexpected given that these measures did not correlate to other risk factors. The only paper investigating PBR-hf and MVHS

did not report on gender difference [18]. Thus, further studies are needed to confirm our findings. In the paper from Wadowski et al [25] there was no statistically significant difference between microcirculatory parameters between men and women with type 1 diabetes, in line with our finding for PBR.

## Associations between endothelial glycocalyx dimension and levels of albuminuria

We believe that the sublingual glycocalyx is representative for e.g. the kidney glycocalyx as the effect on the microvascular vessels is considered a systemic effect [28]. Therefore, it was unexpected that, we were unable to demonstrate a difference in level of PBR, PBR-hf or MVHS between participants with normo-, micro- or macroalbuminuria or with different levels of eGFR. The literature covering this possible association is sparse. Nieuwdorp et al. described in 2006 that the systemic glycocalyx dimension was lower in persons with type 1 diabetes and microalbuminuria (n = 7) compared to persons without microalbuminuria (n = 7) [9]. The study did not use the GlycoCheck system, but orthogonal polarization spectral microscopy and a tracer method (using Dextran) was applied to estimate the glycocalyx and the glycocalyx volume, respectively, this method is not applicable in clinical practice. Furthermore, the sample size was small compared to ours and no information on the difference between persons with micro- and macroalbuminuria was presented. The use of different methods to estimate the glycocalyx might partly explain the conflicting results between the study by Nieuwdorp and ours. In contrast, the study by Amraoui et al. showed that PBR (measured with the GlycoCheck system) was similar in persons with microalbuminuria compared to those without albuminuria, however no information on whether or not these participants had diabetes was provided [16]. Since albuminuria is a sign of damage to the kidneys and a microvascular complication to diabetes, impairment of the endothelial glycocalyx was to be expected. As measurements obtained with the GlycoCheck system did not show any difference in the glycocalyx dimension between normo- micro- and macroalbuminuria this method might not be useful in this context.

The added value provided by estimation of PBR-hf and MVHS has only been reported in an observational study evaluating the sublingual microcirculation in persons with dengue fever [18]. In this study no significant differences in PBR-hf or MVHS between the persons with dengue fever and those with other febrile illness could be demonstrated. However, PBR-hf was higher and MVHS was lower in persons with dengue fever and more severe plasma leakage during the critical phase. In our study we only found a higher PBR-hf and a lower MVHS in females and no relation to other risk factors. Thus, the use of these two new measures did not improve cardio-renal risk evaluation in this population.

## Strengths and limitations

The strengths of our study include the well-defined cohort representing a broad segment of the population with type 1 diabetes exhibiting all stages of albuminuria. We recently investigated the importance of examination conditions in a randomized, controlled study by Eickhoff et al. [6] In this study the reproducibility and influence of examination conditions on measurements with the GlycoCheck system was studied. The study showed that measurements with the GlycoCheck system had a moderate reproducibility, which improved considerably with multiple measurements and had a small day-to-day variability. Moreover, smoking, meal and coffee intake had effects on the GlycoCheck measurements for up to three hours, and abstinence for at least three hours was recommended. Our study adhered to these recommendations. Limitations include the period with technical issues related to the camera. The measurements were not valid

in 15.5% of the persons with type 1 diabetes. As data were missing due to a defect equipment (technical difficulties) and not related to clinical characteristics there was no selection bias and data were missing at random. Moreover, even though a post-hoc sample size analysis suggested a sufficient power of the present analyses, data were originally collected for another primary purpose and the likelihood of type 2 errors cannot be excluded.

### Potential future clinical applications

The lack of association in a cross-sectional study does not rule out a possible predictive value of the glycocalyx dimensions since this design provides a snapshot at a single time point and therefore cannot provide cause and effect relationships between the glycocalyx dimensions and future events. Thus, it remains to be investigated if the glycocalyx measures are predictive of future cardiac and kidney events, and whether they can be used as endpoints in clinical studies with cardio-renal interventions. Moreover, the lack of an association between PBR and MVHS, demonstrating that the two measures may represent different aspects of microvascular damage, could be beneficial in the independent predictive value of these parameter for kidney disease and this remains to be investigated in future prospective studies.

A benefit of measurements with the GlycoCheck system is that it is of low cost and a non-invasive method, without any radiation or discomfort for the persons involved and easy to use.

### Conclusion

In conclusion, we demonstrated that the endothelial glycocalyx dimension assessed with the GlycoCheck system was impaired in persons with type 1 diabetes as compared to healthy controls. Findings were in context with previous studies and follows the hypothesis, that a hyper-glycemic state reduces the size of the glycocalyx size. We found no association between the endothelial glycocalyx dimension and cardio-renal risk factors, except from a higher PBR-hf and a lower MVHS in females. We found no association between the endothelial glycocalyx dimension and the level of albuminuria among persons with type 1 diabetes. Thus, the use of the GlycoCheck system in persons with type 1 diabetes might not contribute to cardiovascular or renal risk stratification. However, larger, dedicated, prospective studies employing the GlycoCheck system are warranted before discarding the method.

### Supporting information

**S1 Questionnaire. Questionnaire regarding PROTON in English.**
(DOCX)

**S2 Questionnaire. Questionnaire regarding PROTON in original language.**
(DOCX)

### Acknowledgments

The authors would like to thank all the participants. We thank T. Ragnholm Juhl, B. Ruud Jensen, J. Herman, A. Lundgaard and L. Jelstrup from Steno Diabetes Center Copenhagen for technical assistance.

### Author Contributions

**Conceptualization:** Tine Willum Hansen, Peter Rossing.

**Data curation:** Signe Abitz Winther, Tine Willum Hansen.

**Formal analysis:** Hanan Amadid.

**Funding acquisition:** Peter Rossing.

**Methodology:** Hanan Amadid, Tine Willum Hansen, Peter Rossing.

**Project administration:** Tine Willum Hansen, Peter Rossing.

**Supervision:** Marie Frimodt-Møller, Frederik Persson, Tine Willum Hansen, Peter Rossing.

**Validation:** Peter Rossing.

**Visualization:** Tine Willum Hansen, Peter Rossing.

**Writing – original draft:** Elisabeth Buur Stougaard.

**Writing – review & editing:** Signe Abitz Winther, Hanan Amadid, Marie Frimodt-Møller, Frederik Persson, Tine Willum Hansen, Peter Rossing.

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
