## [Decision Letter · Decision Letter 0]

26 May 2021

PONE-D-21-11045

To the Editor

Endothelial glycocalyx and cardio-renal risk factors in type 1 diabetes

PLOS ONE

Dear Dr. Stougaard,

Thank you for submitting your manuscript to PLOS ONE. After careful consideration, we feel that it has merit but does not fully meet PLOS ONE’s publication criteria as it currently stands. Therefore, we invite you to submit a revised version of the manuscript that addresses the points raised during the review process.

We look forward to receiving your revised manuscript.

Kind regards,

Harald Mischak

Academic Editor

PLOS ONE

Journal Requirements:

3. In your Methods section, please provide additional information about the participant recruitment method and the demographic details of your participants. Please ensure you have provided sufficient details to replicate the analyses such as:

- the recruitment date range (month and year)

- a statement as to whether your sample can be considered representative of a larger population.

"I have read the journal´s policy and the authors of this manuscript have the following competing interests:

[Peter Rossing (PR) has received research grants from AstraZeneca and Novo Nordisk. He has received lecture and/or consultancy fees (to his institution) from Astellas, AstraZeneca, Bayer, Boehringer Ingelheim, Eli Lilly, Gilead, Merck, Mundipharma, Novo Nordisk and Sanofi Aventis.

Frederik Persson (FP) reports having received research grants from AstraZeneca, Novo Nordisk and Novartis and lecture fees from Novartis, Eli Lilly, MSD, AstraZeneca, Sanofi, Novo Nordisk and Boehringer Ingelheim and having served as a consultant for Astra Zeneca, Bayer, Amgen, Novo Nordisk and MSD]

"

5. We noted in your submission details that a portion of your manuscript may have been presented or published elsewhere.

"Fig. 1 have been published previously in PLOSONE by the same authors.

"Assessment of the sublingual microcirculation

with the GlycoCheck system: Reproducibility

and examination conditions"

6. Please ensure that you include a title page within your main document. You should list all authors and all affiliations as per our author instructions and clearly indicate the corresponding author.

7. Please amend either the title on the online submission form (via Edit Submission) or the title in the manuscript so that they are identical.

Additional Editor Comments:

As you can see from the reviewer comments, the manuscript does have substantial shortcomings, including typographical errors, and apparently the reproduction of a figure that was presented in a previous manuscript. The results are mostly confirmatory, however, the study was appropriately performed and the lack of novelty does not preclude publication in PLOS ONE. As such, I invite you to submit a substantially revised version. If you decide to revise, please pay attention to details, avoid/correct typographical errrors, etc.

Reviewers' comments:

Reviewer's Responses to Questions

**Comments to the Author**

1. Is the manuscript technically sound, and do the data support the conclusions?

Reviewer #1: No

Reviewer #2: Yes

2. Has the statistical analysis been performed appropriately and rigorously? 

Reviewer #1: Yes

Reviewer #2: Yes

3. Have the authors made all data underlying the findings in their manuscript fully available?

Reviewer #1: Yes

Reviewer #2: Yes

4. Is the manuscript presented in an intelligible fashion and written in standard English?

Reviewer #1: No

Reviewer #2: Yes

5. Review Comments to the Author

Reviewer #1: Dr Buur Stougaard and co-workers investigate the association between glycocalyx alterations and cardio-renal risk factors in patients with type 1 diabetes. The paper addresses an interesting topic, but the manuscript has several shortcomings. The following concerns are for the authors’ consideration:

- The relevance of the findings is reduced by the fact that most of the knowledge presented in the paper has already been reported and the results are only confirmatory in nature and do not provide further insights. It is well known that diabetes is associated with endothelial glycocalyx damage and several studies have investigated this concept using the GlycoCheck system. A previous study has shown that in patients with type diabetes 1 there was an inverse correlation between PBR and creatinine but microvascular parameters did not correlate with eGFR. In the same study, there was no statistically significant difference between microcirculatory parameters in men and women with type 1 diabetes (Wadosky 2020). Moreover, Nieuwdorp et al. used orthogonal polarization spectral imaging in patients with type 1 diabetes, observing an association between glycocalyx impairment and microalbuminuria (Nieuwdorp et al., 2006). The authors should consider and comment on their results in light of these data and discuss these discrepancies in greater depth in the Discussion.

Minor points:

- There are several mistakes throughout the text. The GlucoCheck device cited in the Abstract and in the Discussion should be GlycoCheck. Please correct throughout.

- The supporting information comes from a previous paper by the same authors. Is not clear why they used this material instead to only referring to it in in the Methods section, in the text.

- Figure 1 is the same figure that was already published in their previous study. Please replace with another graphical representation or capillaries image or cite it as a reference.

- The Discussion seems to simply repeat the findings that the authors have already presented. The authors should write a more critical Discussion and better interpret their findings, including a discussion regarding potential future clinical developments in this field.

- The figures and tables should be revised. They are difficult to understand and not entirely correct. Specifically, PBR, PBR-HF and MVHS measurements of healthy controls and diabetic patients should be grouped in the same figure or named Figure 2, 3 and 4. The same correction is necessary for Figures 3 and 4. The tables should be visible on the same page, to make it easier to interpret the data.

- There are several mistakes throughout the text. On page 8, line 155, please change “Microvasular” to “Microvascular”.

Reviewer #2: The authors examined the associations between a novel measure of glycocalyx dimension of sublingual endothelium (PBR), Microvascular Health Score (MVHS) and cardio-renal risk factors in T1DM and control subjects. The study was a prespecified analysis of secondary endpoints from a previous study. The found an impaired PBR in T1DM vs controls and higher PBR-hf in females, but no correlation between PBR and albuminuria or cardio-renal risk factors in T1DM.

The subject is important, and the technology novel and potentially interesting. The study is well conducted and presented.

MAJOR COMMENTS:

1. I think that the practical execution of this novel technique should be briefly described in greater detail in this paper, in order to enable the reader to understand the methodology from the present artivcle without reading the referenced papers in detail.

2. The main result was a lack of association between PBR and cardio-renal risk factors/albuminuria. Even though a post-hoc sample size analysis suggested a sufficient power of the present analyses, data were originally collected for another primary purpose. Can a type 2 error be excluded?

3. How do the authors explain the gender difference in PBR-hf (does not seem to correlate to risk factors?)? Is this known from previous studies?

4. A lack of association in a cross-sectional study does not rule out a possible predictive value of PBR for the endpoints studied.

5. The lack of association between PBR and MVHS could actually be beneficial in the independent predictive value of this parameter for kidney disease, given the fact that this could be demonstrated (which it unfortunately could not in the present study).

6. Is the sublingual glycocalyx representative for e.g. the kidney glycocalyx?

7. Do the authors believe that the present study should be followed by a larger, dedicated, prospective study employing a similar methodology before discarding the method?

6. PLOS authors have the option to publish the peer review history of their article (what does this mean?). If published, this will include your full peer review and any attached files.

Reviewer #1: No

Reviewer #2: No

---

## [Author Response · Author response to Decision Letter 0]

15 Jun 2021

To the Editor,

Thank you very much for the opportunity to submit a revised version of our manuscript entitled " Endothelial glycocalyx and cardio-renal risk factors in type 1 diabetes”.

We have revised the manuscript in response to the comments from the in-house editor and the reviewers. The suggestions improved the quality of our manuscript considerably. All changes are highlighted in the manuscript and we have further enclosed an unmarked version of our revised paper and a point-to-point response to the reviewer’s comments.

We hope that with these clarifications and alterations our manuscript will be acceptable for publication. However, we remain prepared to revise our manuscript further, should this be required. 

We are looking forward to hearing from you.

Sincerely and on behalf of all authors,

Elisabeth Buur Stougaard

 

We have ensured that our manuscript meets the PLOS ONE´s style requirements.

Please include additional information regarding the survey or questionnaire used in the study and ensure that you have provided sufficient details that others could replicate the analyses. For instance, if you developed a questionnaire as part of this study and it is not under a copyright more restrictive than CC-BY, please include a copy, in both the original language and English, as Supporting Information.

We have included the questionnaire, in both the original language and in English, as Supporting Information.

In your Methods section, please provide additional information about the participant recruitment method and the demographic details of your participants. Please ensure you have provided sufficient details to replicate the analyses such as: 

As requested, we have added the following to the revised Methods (Page 5; lines 93-98): “Subjects with type 1 diabetes were recruited from Steno Diabetes Center Copenhagen and identified through our electronic medical records. Potentially suitable participants were in writing given the offer to participate. Non-responders were contacted by telephone and given a renewed offer to participate in the study. Healthy control subjects were recruited from newspaper advertisement, where they were encouraged to contact the responsible investigator for more information. If after the telephone conversation the subject was still interested, detailed information was sent for further review”.

- the recruitment date range (month and year) 

We have added the following to the revised Methods (Page 5; line 92): “April 2016 to December 2017”.

- a statement as to whether your sample can be considered representative of a larger population. 

We have added the following to the revised Methods (Page 6; lines 104-109): “Participants included in the current study were recruited from a pool of approximately 3,500 persons with type 1 diabetes attending the outpatient clinic at Steno Diabetes Center Copenhagen. Thus, almost 7% of persons followed-up at Steno Diabetes Center Copenhagen were investigated, representing a broad segment of the Steno population, which covers an unselected population of adults with type 1 diabetes in the capital region of Denmark.”

Thank you for stating the following in the Competing Interests section: "I have read the journal´s policy and the authors of this manuscript have the following competing interests: [Peter Rossing (PR) has received research grants from AstraZeneca and Novo Nordisk. He has received lecture and/or consultancy fees (to his institution) from Astellas, AstraZeneca, Bayer, Boehringer Ingelheim, Eli Lilly, Gilead, Merck, Mundipharma, Novo Nordisk and Sanofi Aventis. Frederik Persson (FP) reports having received research grants from AstraZeneca, Novo Nordisk and Novartis and lecture fees from Novartis, Eli Lilly, MSD, AstraZeneca, Sanofi, Novo Nordisk and Boehringer Ingelheim and having served as a consultant for Astra Zeneca, Bayer, Amgen, Novo Nordisk and MSD]" Please confirm that this does not alter your adherence to all PLOS ONE policies on sharing data and materials, by including the following statement: "This does not alter our adherence to PLOS ONE policies on sharing data and materials.” (as detailed online in our guide for authors http://journals.plos.org/plosone/s/competing-interests). If there are restrictions on sharing of data and/or materials, please state these. Please note that we cannot proceed with consideration of your article until this information has been declared. Please include your updated Competing Interests statement in your cover letter; we will change the online submission form on your behalf.

Thank you for pointing this out. We have included our updated Competing interests’ statement in the Cover Letter: "I have read the journal´s policy and the authors of this manuscript have the following competing interests: [Peter Rossing (PR) has received research grants from AstraZeneca and Novo Nordisk. He has received lecture and/or consultancy fees (to his institution) from Astellas, AstraZeneca, Bayer, Boehringer Ingelheim, Eli Lilly, Gilead, Merck, Mundipharma, Novo Nordisk and Sanofi Aventis. Frederik Persson (FP) reports having received research grants from AstraZeneca, Novo Nordisk and Novartis and lecture fees from Novartis, Eli Lilly, MSD, AstraZeneca, Sanofi, Novo Nordisk and Boehringer Ingelheim and having served as a consultant for Astra Zeneca, Bayer, Amgen, Novo Nordisk and MSD]. This does not alter our adherence to PLOS ONE policies on sharing data and materials."

We noted in your submission details that a portion of your manuscript may have been presented or published elsewhere. "Fig. 1 have been published previously in PLOSONE by the same authors. "Assessment of the sublingual microcirculation with the GlycoCheck system: Reproducibility and examination conditions" Please clarify whether this [conference proceeding or publication] was peer-reviewed and formally published. If this work was previously peer-reviewed and published, in the cover letter please provide the reason that this work does not constitute dual publication and should be included in the current manuscript. Thank you for this comment. Figure 1 is a representation of the Glycocheck method itself and therefore not a dual publication 

Reviewer #1: Dr Buur Stougaard and co-workers investigate the association between glycocalyx alterations and cardio-renal risk factors in patients with type 1 diabetes. The paper addresses an interesting topic, but the manuscript has several shortcomings. The following concerns are for the authors’ consideration:

- The relevance of the findings is reduced by the fact that most of the knowledge presented in the paper has already been reported and the results are only confirmatory in nature and do not provide further insights. It is well known that diabetes is associated with endothelial glycocalyx damage and several studies have investigated this concept using the GlycoCheck system. 

Thank you for this comment. We have highlighted the novelty to the revised Introduction (Page 5; lines 76-78): “None of these studies have been performed in a cohort solely of persons with type 1 diabetes, as in our study, and with a sample size as large as ours.

Page 5; lines 80-82): “The importance of these new measures has only been sparsely investigated and, to our knowledge, described only in one study focusing on dengue and other febrile illness [Lam et al. 2020]”. Thus, the impact of these measures in persons with diabetes is unknown.

A previous study has shown that in patients with type diabetes 1 there was an inverse correlation between PBR and creatinine, but microvascular parameters did not correlate with eGFR. In the same study, there was no statistically significant difference between microcirculatory parameters in men and women with type 1 diabetes (Wadosky 2020). 

We have, in accordance with this important comment, expanded the revised Discussion (page 19; lines 335-338: “The study also showed an inverse correlation between PBR and creatinine, but microvascular parameters (PBR, red blood cells filling percentage, perfused and total capillary density) did not correlate with eGFR [Wadowski et al., 2020]. We were also unable to show a correlation between endothelial glycocalyx dimension and eGFR”.

Page 19; lines 339-343: “The gender difference in PBR-hf and MVHS was quite unexpected given that these measures did not correlate to other risk factors. The only paper investigating PBR-hf and MVHS did not report on gender difference [Lam et al., 2020]. Thus, further studies are needed to confirm our findings. In the paper from Wadowski et al [Wadowski et al., 2020] there was no statistically significant difference between microcirculatory parameters between men and women with type 1 diabetes, in line with our finding for PBR”.

Moreover, Nieuwdorp et al. used orthogonal polarization spectral imaging in patients with type 1 diabetes, observing an association between glycocalyx impairment and microalbuminuria (Nieuwdorp et al., 2006). 

Thank you for this relevant comment. We have expanded the revised Discussion (page 20; lines 350-357: “Nieuwdorp et al. described in 2006 that the systemic glycocalyx dimension was lower in persons with type 1 diabetes and microalbuminuria (n=7) compared to persons without microalbuminuria (n=7) [Nieuwdorp et al., 2006]. The study did not use the GlycoCheck system, but orthogonal polarization spectral microscopy and a tracer method (using Dextran) was applied to estimate the glycocalyx and the glycocalyx volume, respectively, this method is not applicable in clinical practice. Furthermore, the sample size was small compared to ours and no information on the difference between persons with micro- and macroalbuminuria was presented. The use of different methods to estimate the glycocalyx might partly explain the conflicting results between the study by Nieuwdorp and ours”.

The authors should consider and comment on their results in light of these data and discuss these discrepancies in greater depth in the Discussion.

Thank you for this valuable suggestion, we have expanded the Discussion on discrepancies with the text included to the comment above and on page 20; lines 357-360: “In contrast, the study by Amraoui et al. showed that PBR (measured with the GlycoCheck system) was similar in persons with microalbuminuria compared to those without albuminuria, however no information on whether or not these participants had diabetes was provided.

Minor points:

- There are several mistakes throughout the text. The GlucoCheck device cited in the Abstract and in the Discussion should be GlycoCheck. Please correct throughout.

Thank you very much for pointing this out. We apologize for the mistakes and have corrected throughout.

- The supporting information comes from a previous paper by the same authors. Is not clear why they used this material instead to only referring to it in in the Methods section, in the text.

As suggested, we have expanded the revised Methods section with supporting information:

Page 7-8; lines 139-144: “In short, the system consists of a small handheld video camera connected to a computer with the GlykoCheck software. The microscopy was performed with the participant sitting on a chair. The sublingual capillaries were visualized using a SDF video microscope (Capiscope handheld, KK Research technology Ltd), which uses green light emitting diodes to detect the haemoglobin of passing red blood cells (RBC), and the dimensions of the glycocalyx are then estimated with the integrated software (GlycoCheck™, Maastricht, The Netherlands)”.

Page 8; lines 149-154: “For each vascular segment, the dynamic lateral position of RBCs (per RBC column width) is then calculated. The cumulative distribution is calculated from the intensity profiles of the dispersal of RBC column widths and the median RBC column width is determined. The PBR is then defined as the distance between RBC column width and perfused diameter. Next, the calculated PBR values, classified along with their corresponding RBC column width between 5–25 µm, are averaged and a single PBR value is provided for each person”.

- Figure 1 is the same figure that was already published in their previous study. Please replace with another graphical representation or capillaries image or cite it as a reference. 

We have as suggested cited it as a reference.

- The Discussion seems to simply repeat the findings that the authors have already presented. The authors should write a more critical Discussion and better interpret their findings, including a discussion regarding potential future clinical developments in this field. 

Thank you for this important comment. We have expanded the revised Discussion and included a new section on potential future clinical applications (page 21; lines 386-396): 

“Potential future clinical applications

The lack of association in a cross-sectional study does not rule out a possible predictive value of the glycocalyx dimensions, and it remains to be investigated if the glycocalyx measures are predictive of future cardiac and kidney events, and whether they can be used as endpoints in clinical studies with cardio-renal interventions. Moreover, the lack of association between PBR and MVHS could be beneficial in the independent predictive value of these parameter for kidney disease and this remains to be investigated in future prospective studies. 

A benefit of measurements with the GlycoCheck system is that it is of low cost and a non-invasive method, without any radiation or discomfort for the persons involved and easy to use”.

- The figures and tables should be revised. They are difficult to understand and not entirely correct. Specifically, PBR, PBR-HF and MVHS measurements of healthy controls and diabetic patients should be grouped in the same figure or named Figure 2, 3 and 4. The same correction is necessary for Figures 3 and 4. 

We have revised the figures and tables in accordance with this helpful comment.

The tables should be visible on the same page, to make it easier to interpret the data.

We agree and have formatted the tables accordantly.

- There are several mistakes throughout the text. On page 8, line 155, please change “Microvasular” to “Microvascular”.

We apologize and have corrected.

Reviewer #2: The authors examined the associations between a novel measure of glycocalyx dimension of sublingual endothelium (PBR), Microvascular Health Score (MVHS) and cardio-renal risk factors in T1DM and control subjects. The study was a prespecified analysis of secondary endpoints from a previous study. The found an impaired PBR in T1DM vs controls and higher PBR-hf in females, but no correlation between PBR and albuminuria or cardio-renal risk factors in T1DM.

The subject is important, and the technology novel and potentially interesting. The study is well conducted and presented.

MAJOR COMMENTS:

1. I think that the practical execution of this novel technique should be briefly described in greater detail in this paper, in order to enable the reader to understand the methodology from the present article without reading the referenced papers in detail. 

Thank you for this relevant comment. We have included a more detailed description of the technique in the revised Methods:

Page 7-8; lines 139-144: “In short, the system consists of a small handheld video camera connected to a computer with the GlykoCheck software. The microscopy was performed with the participant sitting on a chair. The sublingual capillaries were visualized using a SDF video microscope (Capiscope handheld, KK Research technology Ltd), which uses green light emitting diodes to detect the haemoglobin of passing red blood cells (RBC), and the dimensions of the glycocalyx are then estimated with the integrated software (GlycoCheck™, Maastricht, The Netherlands)”.

Page 8; lines 149-154: “For each vascular segment, the dynamic lateral position of RBCs (per RBC column width) is then calculated. The cumulative distribution is calculated from the intensity profiles of the dispersal of RBC column widths and the median RBC column width is determined. The PBR is then defined as the distance between RBC column width and perfused diameter. Next, the calculated PBR values, classified along with their corresponding RBC column width between 5–25 µm, are averaged and a single PBR value is provided for each person”.

2. The main result was a lack of association between PBR and cardio-renal risk factors/albuminuria. Even though a post-hoc sample size analysis suggested a sufficient power of the present analyses, data were originally collected for another primary purpose. Can a type 2 error be excluded? 

The likelihood of a type 2 error cannot be excluded, and we have revised the Discussion in accordance with your helpful comment:

Page 21; lines 383-385: “Moreover, even though a post-hoc sample size analysis suggested a sufficient power of the present analyses, data were originally collected for another primary purpose and the likelihood of type 2 errors cannot be excluded”.

3. How do the authors explain the gender difference in PBR-hf (does not seem to correlate to risk factors?)? Is this known from previous studies? 

In line with this important comment we have added to the following to the revised Discussion

Page 19; lines 339-343: “The gender difference in PBR-hf and MVHS was quite unexpected given that these measures did not correlate to other risk factors. The only paper investigating PBR-hf and MVHS did not report on gender difference [Lam et al., 2020]. Thus, further studies are needed to confirm our findings. In the paper from Wadowski et al [Wadosky et al., 2020] there was no statistically significant difference between microcirculatory parameters between men and women with type 1 diabetes, in line with our finding for PBR”.

4. A lack of association in a cross-sectional study does not rule out a possible predictive value of PBR for the endpoints studied. 

We fully agree and have revised the Discussion accordantly (page 21; lines 387-391): “The lack of association in a cross-sectional study does not rule out a possible predictive value of the glycocalyx dimensions since this design provides a snapshot at a single time point and therefore cannot provide cause and effect relationships between the glycocalyx dimensions and future events. Thus, and it remains to be investigated if the glycocalyx measures are predictive of future cardiac and kidney events, and whether they can be used as endpoints in clinical studies with cardio-renal interventions”. 

5. The lack of association between PBR and MVHS could actually be beneficial in the independent predictive value of this parameter for kidney disease, given the fact that this could be demonstrated (which it unfortunately could not in the present study). 

Thank you for this helpful comment, we have revised the Discussion accordantly (page 21; lines 391-394): “Moreover, the lack of an association between PBR and MVHS, demonstrating that the two measures may represent different aspects of microvascular damage, could actually be beneficial in the independent predictive value of these parameters for kidney disease and this remains to be investigated in future prospective studies.”

6. Is the sublingual glycocalyx representative for e.g. the kidney glycocalyx? 

This is an important question and we have added to the revised Discussion (page 20; lines 346-347): “We believe that the sublingual glycocalyx is representative for e.g. the kidney glycocalyx as the effect on the microvascular vessels is considered a systemic effect (Deckert T et.al. 1989).” 

7. Do the authors believe that the present study should be followed by a larger, dedicated, prospective study employing a similar methodology before discarding the method? 

Thank you for this relevant question. We believe that larger, dedicated, prospective study employing the methodology is warranted before discarding the method. This is included in the revised Discussion (page 22; lines 406-407): “However, larger, dedicated, prospective studies employing the GlycoCheck system are warranted before discarding the method”.

---

## [Decision Letter · Decision Letter 1]

6 Jul 2021

Endothelial glycocalyx and cardio-renal risk factors in type 1 diabetes

PONE-D-21-11045R1

Dear Dr. Stougaard,

We’re pleased to inform you that your manuscript has been judged scientifically suitable for publication and will be formally accepted for publication once it meets all outstanding technical requirements.

Kind regards,

Harald Mischak

Academic Editor

PLOS ONE

Additional Editor Comments (optional):

Reviewers' comments:

Reviewer's Responses to Questions

**Comments to the Author**

1. If the authors have adequately addressed your comments raised in a previous round of review and you feel that this manuscript is now acceptable for publication, you may indicate that here to bypass the “Comments to the Author” section, enter your conflict of interest statement in the “Confidential to Editor” section, and submit your "Accept" recommendation.

Reviewer #2: All comments have been addressed

Reviewer #3: All comments have been addressed

2. Is the manuscript technically sound, and do the data support the conclusions?

Reviewer #2: Yes

Reviewer #3: Yes

3. Has the statistical analysis been performed appropriately and rigorously? 

Reviewer #2: Yes

Reviewer #3: Yes

4. Have the authors made all data underlying the findings in their manuscript fully available?

Reviewer #2: Yes

Reviewer #3: Yes

5. Is the manuscript presented in an intelligible fashion and written in standard English?

Reviewer #2: Yes

Reviewer #3: Yes

6. Review Comments to the Author

Reviewer #2: The authors have meticulously addressed reviewer comment and have made corresponding changes to the manuscript, which has significantly improved the quality. I have no further suggestions.

Reviewer #3: All reviewers' questions were promptly answers.

There is however one point about data accessibility in the manuscript: the authors should have probably selected "Data cannot be shared publicly because of [XXX]. Data are available from the XXX Institutional Data Access / Ethics Committee (contact via XXX) for researchers who meet the criteria for access to confidential data.". Because it seems that not all data (clinical and about measurements) were revealed in the manuscript, which is understandable because of privacy/ethical issue.

Otherwise, the manuscript is considered carefully revised that I am happy to recommend it for acceptance.

7. PLOS authors have the option to publish the peer review history of their article (what does this mean?). If published, this will include your full peer review and any attached files.

Reviewer #2: No

Reviewer #3: No

---

## [Editor Report · Acceptance letter]

12 Jul 2021

PONE-D-21-11045R1 

Endothelial glycocalyx and cardio-renal risk factors in type 1 diabetes 

Dear Dr. Stougaard:

I'm pleased to inform you that your manuscript has been deemed suitable for publication in PLOS ONE. Congratulations! Your manuscript is now with our production department. 

Kind regards, 

on behalf of

Prof. Harald Mischak 

Academic Editor

PLOS ONE